# Eryptosis in Liver Diseases: Contribution to Anemia and Hypercoagulation

**DOI:** 10.3390/medsci13030125

**Published:** 2025-08-12

**Authors:** Saulesh Kurmangaliyeva, Kristina Baktikulova, Anton Tkachenko, Bibigul Seitkhanova, Liliya Tryfonyuk, Farida Rakhimzhanova, Rustam Yussupov, Kairat Kurmangaliyev

**Affiliations:** 1Department of Microbiology, Virology, and Immunology, NJSC “West Kazakhstan Marat Ospanov Medical University”, 68 Maresyev st, Aktobe 030000, Kazakhstan; saule_cc@mail.ru (S.K.); kairat_121@mail.ru (K.K.); 2BIOCEV, First Faculty of Medicine, Charles University, Průmyslová 595, 25250 Vestec, Czech Republic; 3Department of Microbiology, Virology, and Immunology, NJSC “South Kazakhstan Medical Academy”, Al-Farabi sq, Shymkent 160019, Kazakhstan; s.bibigul@smka.kz; 4Institute of Health, National University of Water and Environmental Engineering, 11 Soborna st, 33028 Rivne, Ukraine; liliya_tryfonyuk@yahoo.pl; 5Rivne Regional Clinical Hospital, 78g Kyivska st, 33007 Rivne, Ukraine; 6Department of Microbiology, NJSC “Semey Medical University”, 103 Abay st, Semey 071400, Kazakhstan; farida.rakhimzhanova@smu.edu.kz; 7Department of Microbiology, Virology, Republic NJSC “Kazakh National Medical University Named after S.D. Asfendiyarov”, 94 Tole Bi, Almaty 005000, Kazakhstan; r.yussupov@kaznmu.kz

**Keywords:** bile acids, bilirubin, chronic liver disease, eryptosis, non-alcoholic fatty liver disease, regulated cell death

## Abstract

Eryptosis is a type of regulated cell death of mature erythrocytes characterized by excessive Ca^2+^ accumulation followed by phosphatidylserine externalization. Eryptosis facilitates erythrophagocytosis resulting in eradication of damaged erythrocytes, which maintains the population of healthy erythrocytes in blood. Over recent years, a wide array of diseases has been reported to be linked to accelerated eryptosis, which leads to anemia. A growing number of studies furnish evidence that eryptosis is implicated in the pathogenesis of liver diseases. Herein, we summarize the current knowledge of eryptosis signaling, its physiological role, and the impact of eryptosis on anemia and hypercoagulation. In this article, upon systemically analyzing the PubMed-indexed publications, we also provide a comprehensive overview of the role of eryptosis in the spectrum of hepatic diseases, its contribution to the development of complications in liver pathology, metabolites (bilirubin, bile acids, etc.) that might trigger eryptosis in liver diseases, and eryptosis-inducing liver disease medications. Eryptosis in liver diseases contributes to anemia, hypercoagulation, and endothelial damage (via ferroptosis of endothelial cells). Treatment-associated anemia in liver diseases might be at least partly attributed to drug-induced eryptosis. Ultimately, we analyze the concept of inhibiting eryptosis pharmaceutically to prevent eryptosis-associated anemia and thrombosis in liver diseases.

## 1. Introduction

Liver is a key nexus for a variety of physiological functions, playing a central role in carbohydrate, lipid, and protein metabolism, detoxification of xenobiotics, bile formation, production of acute phase proteins, as well as pro- and anti-fibrinolytic proteins, storage of fat-soluble vitamins, heme breakdown, regulation of systemic iron homeostasis, production of the bulk of blood plasma proteins, synthesis of the majority of blood coagulation factors, etc. [1,2,3,4]. In addition, the liver is fundamental for the proper immune response, possessing a large number of phagocytes such as Kupffer cells that represent over one third of non-parenchymatous hepatic cells. This supports the role of the liver as the major barrier that clears various pathogens from the blood [5]. The liver has long been considered to perform endocrine functions, which include one of the steps of vitamin D (cholecalciferol) hydroxylation (25-hydroxylation to form 25-hydroxycholecalciferol) to produce calcitriol (1,25-dihydroxycholecalciferol), a Ca^2+^-regulating hormone [6], and production of insulin-like growth factors 1 and 2 as a response to growth hormone [7]. The liver is also a site for deiodination of thyroid hormones, which occurs by hepatic deiodinases [8]. Moreover, recent advances in omics-based studies have resulted in the emergence of the concept of hepatokines, which are liver-derived biologically active proteins released into the bloodstream. Currently, over 20 distinct hepatokines, including fetuin-A, angiopoietin-like 3 (ANGPTL3), fibroblast growth factor 21 (FGF21), sex hormone-binding globulin (SHBG), etc., have been described, and accumulating evidence points to their dysregulation in liver pathologies such as non-alcoholic fatty liver disease (NAFLD) [9,10]. Additionally, liver cells express functionally active non-coding RNAs, including microRNAs (miRs), long non-coding RNAs (lncRNAs), and circular RNAs (circRNAs), which can be secreted and act as regulatory molecules locally or distantly [11]. Since the liver elicits such highly pleiotropic activities, its dysfunction is linked to multiple detrimental effects, which manifest if liver diseases emerge.

Liver diseases remain a significant medical and social burden responsible for over 2 million deaths globally, which accounts for 4% of all deaths [12]. In particular, chronic liver disease (CLD), which encompasses a long-lasting hepatic inflammation (over 6 months) resulting in damage and altered regeneration of the parenchyma, eventually culminating in fibrosis and cirrhosis [13], comprises NAFLD currently known as metabolic dysfunction-associated steatotic liver disease (MASLD) [14], alcohol-associated liver disease (ALD), chronic viral hepatitis, autoimmune hepatitis, genetic disorders (alpha-1 antitrypsin deficiency, hereditary hemochromatosis, or Wilson disease), or drug-induced liver injury [15,16]. In the structure of CLD, NAFLD prevails and accounts for over 60% of cases. It is followed by hepatitis B virus (HBV) and hepatitis C virus (HCV) infections, as well as ALD [17]. A wide array of conditions, such as portal hypertension, variceal bleeding, portopulmonary hypertension, ascites, peritonitis, hepatopulmonary or hepatorenal syndrome, as well as hepatic encephalopathy, have been reported to be complications of CLD [18]. Of note, hematological complications are quite common for CLD and comprise thrombocytopenia, leukopenia, elevation of circulating levels of von Willebrand factor, reduced synthesis of liver-derived blood coagulation factors, or the imbalance between pro- and anti-fibrinolytic proteins [19,20]. Importantly, anemia has been reported to develop in approximately 75% of patients with CLD [21,22]. The etiology of this anemia is indeed diverse. For instance, iron deficiency anemia in CLD develops as a result of chronic gastrointestinal hemorrhages primarily due to portal hypertensive gastropathy and gastric antral vascular ectasia [23] or impaired iron homeostasis [22]. Anemia in ALD can be linked to folic acid (vitamin B_9_) or cyanocobalamin (vitamin B_12_) insufficiency related to alcohol abuse or malnutrition [24]. Hepatitis-associated aplastic anemia is characterized by pancytopenia due to the failure of the bone marrow to produce new blood cells [25]. Notably, anemia in CLD might be associated with excessive destruction of erythrocytes, e.g., in the spleen as a result of hypersplenism [22,26]. Although immune hemolytic anemia is rarely observed in CLD, it is still a clearly demonstrated CLD complication [27]. Moreover, hemolytic anemia might develop as a result of antiviral therapy of HCV infection. This treatment-associated secondary anemia is linked to destruction of erythrocytes (hemolysis) [28].

At the same time, hemolysis is not the only possible way for erythrocytes to die. A variety of studies have firmly verified that red blood cells (RBCs) may die in an orderly fashion, a process termed “eryptosis” [29,30,31,32,33,34]. Physiologically, eryptosis accomplishes the functions of preserving the cells from hemolysis and eliminating the damaged RBCs, resembling apoptosis in this way, but eryptosis signaling deviates from that in apoptosis, possessing a certain degree of uniqueness [34,35]. A growing body of evidence suggests that eryptosis is of great physiological and pathophysiological significance. In particular, eryptotic erythrocytes are rapidly cleared from circulation [36], which results in a drop in RBC counts. This eryptosis-associated anemia has become a popular focus of research. Accumulating evidence suggests that eryptosis is a driver of anemia in chronic kidney disease [37]. Likewise, enhanced eryptosis is linked to coagulation disorders and inflammation, especially in kidney diseases [38,39,40]. However, the interplay between eryptosis, inflammation, clotting disorders, and anemia in liver diseases is poorly summarized. In this review article, we highlight the current knowledge of eryptosis in liver diseases, investigating its role in the pathogenesis of liver diseases, in particular, in the development of CLD-associated anemia, triggers of eryptosis in liver diseases, potential diagnostic and prognostic applications of eryptosis markers, and its druggability. Thus, in this review, we primarily aim to assess the impact of eryptosis in liver diseases, emphasizing its contribution to anemia and hypercoagulation. Additionally, our review aims at attracting attention to the role of eryptosis in liver diseases to open new frontiers in research for this RCD pathway of mature erythrocytes.

Herein, we systemically analyzed the available PubMed-indexed English-language publications (up to June 2025) using the following combinations of search keywords: “eryptosis”/“erythrocyte apoptosis”/”RBC apoptosis”/“phosphatidylserine exposure”/“phosphatidylserine externalization” AND “hepatic disease”/“liver disease”/”alcohol-associated liver disease”/“ALD”/”non-alcoholic fatty liver disease”/“NAFLD”/”non-alcoholic steatohepatitis”/“NASH”/”chronic liver disease”/“CLD”/”metabolic dysfunction-associated steatotic liver disease”/“MASLD”/”liver cirrhosis”/”hepatitis”/”liver metabolites”/“liver disease markers”/“liver disease medications”/“liver drugs”.

## 2. Eryptosis: Signaling and Functions

### 2.1. Eryptosis Signaling

Regulated cell death (RCD) pathways play an important role during erythropoiesis, which comprises differentiation of hematopoietic stem cells into mature RBCs. In particular, there is compelling evidence that erythroid precursors can be subjected to intrinsic and extrinsic apoptosis, necroptosis, or ferroptosis. However, maturation of RBCs results in enucleation and clearance from organelles, reducing the diversity of the RBC cell death machinery [33]. Nevertheless, over the last two decades, it has become clear that erythrocytes undergo an orderly executed and tightly controlled RCD referred to as “eryptosis”. Eryptosis was initially described as a cell death triggered by ionomycin, a Ca^2+^ ionophore, in a caspase-independent fashion [30]. As experimental data accumulated, the critical role of calcium ions in eryptosis became firmly established [31,32,36]. Ca^2+^ influx triggers activation of scramblase, one of the three enzymes that are involved in the regulation of the phospholipid asymmetry in RBC membranes [41,42]. Asymmetric distribution of phospholipids with the prevalence of phosphatidylserine (PS) in the cytosolic leaflet is achieved by the action of ATP-dependent flippases that translocate PS and phosphatidylethanolamine from the outer leaflet to the cytosolic one and floppases that rearrange phosphatidylcholine by moving it in the opposite direction [43,44,45]. Ca^2+^-mediated activation of scramblase, which unsystematically moves phospholipids in both directions, interferes with the ordered phospholipid asymmetry, resulting in the translocation of PS onto the surface of RBCs, which is a well-known “eat-me” signal for phagocytes [35,46]. Thus, Ca^2+^ overload ensures disruption of the membrane phospholipid asymmetry in mature RBCs. On top of that, Ca^2+^ entry invokes the opening of K^+^ selective Gardos channels, leading to the leakage of potassium ions, subsequent cellular dehydration, and cell shrinkage, a morphological hallmark of eryptosis [47]. Ca^2+^ signaling stimulates calpain to dismantle the cytoskeleton and to ensure membrane blebbing, another morphological feature of eryptosis [48]. An accumulating body of evidence indicates that Ca^2+^ influx is facilitated by accumulation of ceramide, which can be generated by acid or neutral sphingomyelinase (SMase) through sphingomyelin hydrolysis [49]. Likewise, prostaglandin E_2_ (PGE_2_), generated by cyclooxygenase from arachidonic acid released from phospholipids under the action of phospholipase A2, stimulates Ca^2+^ entry [50].

Besides Ca^2+^ influx, scramblase-mediated PS externalization in mature RBCs can be triggered by reactive oxygen and nitrogen species (ROS and RNS, respectively) [51], caspase-3 [52], or ceramide [53]. In particular, ROS in erythrocytes are generated as a result of hemoglobin autooxidation, Fe^2+^-driven Fenton reaction, by NADPH oxidase, or xanthine oxidoreductase. Moreover, immune cell-derived exogenous ROS may enter erythrocytes as well. It has been reported that ROS induce eryptosis by promoting Ca^2+^ influx (cation channel-driven eryptosis), in a caspase-3-dependent way, and via ceramide production [51]. It can be assumed that ROS-mediated eryptosis might prevent further generation of ROS by eliminating ROS-producing RBCs. Notably, caspases are not critically required for Ca^2+^-dependent cation channel-mediated eryptosis. At the same time, caspase-8 and caspase-3 are activated downstream of Fas signaling, being the key components of Fas/FasL-mediated extrinsic eryptosis [54].

Thus, cation channel-driven, lipid-driven (ceramide), ROS-mediated, caspase-dependent, and death receptor-mediated extrinsic eryptosis types are identified in the recently published eryptosis guidelines on the basis of the key signaling pathways involved [34].

### 2.2. Physiological Functions of Eryptosis

As a general rule, physiological functions of eryptosis have been considered from three major standpoints: the stress response, the regulation of the immune response, and an antimalarial protective mechanism [34].

Over recent years, eryptosis-triggering stress-related alterations in RBCs have been relatively well described. Compelling evidence suggests that eryptosis is induced by metabolic stress linked to severe depletion of ATP [55], hyperosmotic stress [38,41], ion imbalance [56], oxidative stress [37,51], or nitrosative stress [57]. Additionally, a multitude of metabolites or exogenous compounds of various chemical structures can induce eryptosis [32]. It should be noted that stress-associated conditions in erythrocytes trigger PS externalization by either Ca^2+^ overload or modulation of scramblase activity. Notably, it can be assumed that the strength of a stress factor might be important, since there is some evidence that intracellular Ca^2+^ should exceed a certain concentration of approximately 10 µM in order to efficiently trigger phospholipid asymmetry [58]. Importantly, the significant amount of PS on the surface of an erythrocyte promotes the rapid clearance of erythrocytes by erythrophagocytosis, which is orders of magnitude faster compared to senescent RBCs [59]. Thus, the eryptosis-associated removal of stressed erythrocytes is essential to deter hemolysis, an accidental cell death of RBCs linked to detrimental consequences developing from destruction of cell membranes (Figure 1).

Although the field of immunogenic consequences of RBC death is in its infancy, it is clear that dead erythrocytes might release pro-inflammatory damage-associated molecular patterns (DAMPs) like nucleated cells. This fact attracts attention to the role of different RBC death pathways in the regulation of the immune response. Regrettably, this field is underexplored. Nevertheless, it has been widely recognized that erythrocyte-derived DAMPs released as a result of hemolysis like heme or its derivatives promote inflammation serving as ligands for Toll-like receptors, or encouraging the assembly and activation of inflammasomes [60]. Moreover, heme can trigger platelet activation, compromising hemostasis. In addition, heme promotes NETosis, a cell death of neutrophils linked with the formation of neutrophil extracellular traps [61]. Heme-induced ferroptosis is another pro-inflammatory mechanism that mediates heme-associated inflammation-inducing effects of hemolysis, since heme has been shown to induce ferroptosis [62], which is an iron-driven RCD that might amplify inflammation [63]. Moreover, Fortes et al. showed that heme could induce necroptosis in macrophages [64], and necroptosis is a strongly pro-inflammatory regulated necrosis associated with the mixed lineage kinase domain-like pseudokinase (MLKL)-mediated release of DAMPs [65]. Although direct immunogenic effects of eryptosis are still poorly understood, at least indirectly it definitely elicits anti-inflammatory properties by preventing hemolysis-associated heme release.

Moreover, accumulating evidence suggests that eryptosis can be involved in the host defense, in particular, in *Plasmodium* infection. Malaria caused by *Plasmodium* parasites, e.g., *P. falciparum*, has been reported to be linked with enhanced eryptosis of both infected and non-infected erythrocytes. This can lead to a faster clearance of the parasite and prevents invasion of novel RBCs [66].

## 3. Eryptosis in Disease: Eryptosis as a Contributing Factor to Anemia and Thrombosis

Compelling evidence suggests that eryptosis induction is observed in multiple diseases, which has been comprehensively reviewed in a growing number of articles [30,31,32,38]. However, consequences and the targetability of the components of the eryptotic machinery are still underexplored. It is important to note that enhanced eryptosis might be associated with anemia due to removal of the eryptotic RBCs by phagocytes. This process depends on PS externalization, and PS exposed on the surface of eryptotic cells is a major trigger of phagocytosis, which ensures the swift clearance of eryptotic cells [36]. Moreover, a growing body of evidence indicates that eryptosis might contribute to thrombo-occlusive complications in a variety of diseases [67]. Notably, eryptosis-associated PS externalization seems to be a critical factor for eryptosis-associated prothrombotic events. For instance, PS mediates the interaction of eryptotic erythrocytes with platelets through CXCL16 and CD36 [68]. Another mechanism that is involved in thrombosis induction by PS-expressing eryptotic erythrocytes is platelet-independent and includes formation of the prothrombinase complex triggered by PS [69]. This results in the accelerated conversion of prothrombin into thrombin and the subsequent generation of fibrin from fibrinogen. Notably, CXCL16 and CD36 are expressed on the surface of vascular endothelial cells and can be implicated in vascular damage triggered by eryptotic cells [68]. Additionally, endothelial cells in microvasculature have been demonstrated to bind PS-expressing erythrocytes via a functionally active receptor for PS-positive erythrocytes [70]. Notably, inflammation can enforce the negative effect of eryptosis on endothelial function, since this binding is facilitated by tumor necrosis factor alpha (TNF-α), interleukin-1beta (IL-1β), bacterial lipopolysaccharide, in response to hypoxia, and erythrocyte-derived DAMPs [70]. It is important to underscore that although there are some insights into the effects of the interaction between vascular cells and eryptotic erythrocytes, molecular mechanisms that mediate eryptosis-associated thrombosis and endothelial dysfunction are poorly explored. However, it is clear that PS plays a pivotal role in these effects. Nevertheless, it is important to further investigate PS-independent mechanisms that might be involved in supporting pathological consequences of accelerated eryptosis, including in liver diseases.

## 4. Eryptosis and Liver-Derived Metabolites

As outlined above, the liver plays a key role in the regulation of overall metabolism, performing a wide array of metabolism-regulating functions. Thus, liver dysfunction is associated with significant changes in the blood biochemical profile, frequently resulting in elevation of circulating levels of certain metabolites, including bilirubin, bile acids, hepatic enzymes, lipids and lipoproteins, etc. [71,72,73]. Thus, in liver diseases, erythrocytes are exposed to higher concentrations of these metabolites, which might exert erythrotoxicity absent in physiological conditions. In this section, we aim to summarize the currently available data on the effects of liver disease biomarkers on RBC death (summarized in Table 1) to shed light on the pathogenetic factors that might lead to accelerated eryptosis in liver diseases.

Our comprehensive analysis revealed that a variety of metabolites whose blood levels might be elevated in liver diseases could induce eryptosis.

Bilirubin, which is a key product of heme catabolism generated primarily as a result of hemoglobin breakdown, plays a critical diagnostic role in liver diseases, reflecting the degree of liver damage [81]. The impact of bilirubin on RBC viability has been studied for over two decades. In 2002, Brito et al. demonstrated that unconjugated bilirubin was capable of altering the lipid composition of erythrocyte cell membranes promoting elution of phospholipids, which was associated with an increase in the cholesterol/phospholipid ratio. At the same time, the impact of unconjugated bilirubin was linked to alterations in phospholipid membrane asymmetry accompanied by PS externalization [79], a well-known sign of eryptosis. Notably, exposure to high concentrations of unconjugated bilirubin (171 µM) culminated in erythrocyte lysis [82]. Importantly, acidosis worsened unconjugated bilirubin-induced cytotoxicity against erythrocytes [80,82]. Conjugated bilirubin promoted PS externalization, Ca^2+^ overload, and ceramide generation in a dose-dependent manner in vitro [75]. Notably, these findings were confirmed by in vivo studies, demonstrating that bile duct ligation in a murine model resulted in enhanced eryptosis-associated eryptosis with compensatory reticulocyte generation. In vivo, the degree of eryptosis induction was proportional to circulating bilirubin levels [75]. Thus, both conjugated and unconjugated bilirubin promote PS externalization, indicating activation of eryptosis (Figure 2). Lang et al. suggested that 3 mg/dL was a critical threshold of conjugated bilirubin concentration, which led to anemia when exceeded due to eryptosis induction [75]. Taken together, compelling evidence indicates that bilirubin elevation in liver diseases contributes to anemia, at least partly due to eryptosis induction and the subsequent loss of erythrocytes as a result of PS exposure-associated erythrophagocytosis.

Recent advances in bile acid physiology and pathophysiology underscore the important contribution of bile acids to the pathogenesis of liver diseases, and even emphasize the potential of their pharmaceutical targeting for therapy of liver diseases [83,84,85]. Hypercholanemia, an increase in circulating levels of bile acids, has been linked to RBC destruction and anemia for decades. In 1987, cholic, chenodeoxycholic, deoxycholic, and lithocholic acids, as well as their conjugates with glycine or taurine, were shown to promote hemolysis and Ca^2+^ influx. Notably, the effect was the most pronounced in the case of deoxycholic acid [74]. Importantly, bile acid-induced hemolysis was found to be mammalian speciesdependent [86]. Ca^2+^ elevation in erythrocytes following exposure to bile acids allows us to presume that bile acids may induce eryptosis. Indeed, glycochenodesoxycholic and taurochenodesoxycholic acids were shown to induce eryptosis in vitro [77]. Bile acid-induced eryptosis was characterized by PS externalization, cell shrinkage, Ca^2+^ accumulation inside RBCs, and ceramide production [77] (Figure 2). Notably, ATP depletion aggravates erythrotoxicity of bile acids, making erythrocytes with energy deficit more susceptible to bile acid-induced Ca^2+^ overload [74]. Thus, anemia in cholestasis might be attributable to hypercholanemia.

In hepatic diseases such as ALD and NAFLD, saturated fatty acids levels are elevated in blood [87]. Moreover, liver-derived lipids act as regulatory signaling molecules in liver diseases [88]. Alfhili et al. showed that lauric acid, a saturated fatty acid, could trigger eryptosis in a Ca^2+^-dependent manner. Oxidative stress and casein kinase 1α (CK1α) were reported as signaling pathways involved in lauric acid-mediated RBC death [78].

C-reactive protein (CRP) is a well-characterized hepatocyte-derived acute phase protein, which is widely used as an inflammation marker [89]. Importantly, it has been characterized as a biomarker in patients with liver dysfunction [90,91]. Moreover, CRP has emerged as a regulator of MASLD in aged patients [92]. Abed et al. showed that CRP triggered eryptosis via Ca^2+^ influx, promotion of ceramide generation, and activation of caspase-3 [76]. Importantly, correlation between CRP levels and eryptosis were reported for smokers [93]. Although there is some evidence that CRP can induce eryptosis, its linkage with liver diseases is poorly explored.

Thus, there is compelling evidence that at least partly enhanced eryptosis in liver diseases might be attributable the erythrotoxic effects of bilirubin and bile acids. Notably, enhanced degradation of erythrocytes via eryptosis in response to hyperbilirubinemia contributes to accelerated hemoglobin breakdown and subsequent generation of more bilirubin, forming a vicious cycle [94]. However, investigation of eryptosis-inducing capabilities of liver-derived metabolites will gain further importance in order to deepen our understanding of direct eryptosis triggers in liver diseases.

## 5. Eryptosis and Liver Diseases

### 5.1. PS-Displaying Eryptotic Cells Are Cleared by Hepatic Kupffer Cells

Currently, there is a dearth of experimental data elucidating the contribution of eryptosis to liver disease-associated anemia with a limited set of the diseases investigated (Table 2). Enhanced eryptosis verified by PS externalization was shown to occur in rabbits fed a cholesterol-rich and high-fat diet leading to the development of non-alcoholic steatohepatitis (NASH) [95]. Notably, these PS-exposing RBCs were prone to clearance by liver-residing Kupffer cells. This finding is in line with another report supporting sequestration of PS-exposing eryptotic RBCs by Kupffer cells, with the critical importance of stabilin-1 and stabilin-2 expressed in hepatic sinusoidal endothelial cells (HSECs) for this process. Subsequently, the PS-displaying eryptotic cells sequestrated by HSECs in a stabilin-1- and stabilin-2-dependent fashion are presented to Kupffer cells [96].

### 5.2. Eryptosis Is Frequently Enhanced in Liver Diseases

Little is known about eryptosis in specific liver diseases. For instance, anemia in hepatitis B-related acute-on-chronic liver failure (HB-ACLF) was at least partly related to enhanced eryptosis, verified by PS translocation to the outer leaflet and an increase in intracellular Ca^2+^ levels [98]. Analysis of the potential molecular mechanisms revealed that eryptosis induction in HB-ACLF was attributable to oxidative stress [98]. Notably, Mei et al. reported that PS externalization was not higher in patients with chronic hepatitis B and liver cirrhosis, suggesting that eryptosis was not accelerated [98]. On the contrary, Wu et al. reported that PS externalization was higher in patients with cirrhosis compared to healthy volunteers [99]. Notably, the degree of PS exposure was dependent on the class of patients in accordance with the Child–Pugh score, which was used to predict the mortality in liver cirrhosis [99], indicating that eryptosis in cirrhosis was disease severitydependent. PS exposure was higher in erythrocytes treated with ethanol, indicating the contribution of eryptosis to ALD [100]. However, this hypothesis has not yet been tested in vivo. Importantly, a high-fat diet is known to be one of the triggers of NAFLD [101]. At the same time, high-fat diets stimulated PS externalization in RBCs, and hence eryptosis [102,103]. Although there is an obvious trend for increasing the number of published papers on eryptosis, scarce and contradictory data on the links between eryptosis and liver diseases encourage investigation of this topic to shed light on the pathogenesis of anemia and coagulation disorders in patients with liver diseases.

### 5.3. Eryptosis in Liver Diseases Is Triggered by Bilirubin, Bile Acids, Cytokines, and ROS

The factors that might trigger eryptosis in liver diseases are not fully elucidated. Notably, HB-ACLF-derived plasma triggers PS externalization in erythrocytes collected from healthy individuals, suggesting that the blood plasma of these patients contained eryptosis-triggering compounds. According to the authors, this effect might be attributable to circulating bilirubin, bile acids, or pro-inflammatory cytokines [98]. Indeed, Lang et al. demonstrated that induction of eryptosis (PS externalization) in patients (n = 27) with unspecified liver diseases was directly proportional to circulating bilirubin levels, and plasma of patients with liver diseases triggered eryptosis in RBCs derived from healthy volunteers [75]. Moreover, the in vitro studies outlined above clearly support the hypothesis on the contribution of circulating bilirubin and bile acids to eryptosis induction. Abundant evidence indicates that oxidative stress is considerably implicated in liver pathophysiology [104]. Imbalance between prooxidants and antioxidants with excessive formation of ROS and RNS is an important trigger of eryptosis [51]. Immune dysfunction associated with alterations in pro-inflammatory cytokine production is common for liver cirrhosis [105], ALD and NAFLD [106,107], hepatocellular carcinoma [108], and other liver diseases. There is accumulating evidence that pro-inflammatory cytokines might induce eryptosis, linking eryptosis and inflammation. For instance, IL-1 and IL-6 induce eryptosis in murine erythrocytes, promoting PS exposure, typical morphological alterations like cell shrinkage, and ceramide formation [109]. Likewise, eryptosis-like morphological changes were observed in response to IL-8 [110]. Additionally, eryptosis was stimulated by bioymifi, a TNF-related apoptosis-induced ligand (TRAIL) mimetic [111]. Taken together, cytokines might be important triggers of eryptosis in liver diseases. It is worth mentioning that no correlation between eryptosis and the major liver biomarkers besides bilirubin (e.g., alanine aminotransferase, ALT; aspartate aminotransferase, AST; γ-glutamyltransferase, γ-GT, etc.) was found in patients with liver disorders [75]. Thus, eryptosis induction in liver diseases might be associated with elevation of circulating bilirubin and bile acids, as well as the development of oxidative stress and the action of pro-inflammatory cytokines. However, our understanding of the pro-eryptotic factors in liver diseases remains elusive. The search for novel triggers of eryptosis in liver diseases is a promising research direction that might deepen our knowledge of the pathogenesis of liver diseases and shed light on the factors contributing to anemia development.

### 5.4. Eryptosis in Liver Diseases Leads to Anemia

Indeed, anemia is a commonly recognized outcome of enhanced eryptosis [34,41]. Anemia develops against the background of the rapid clearance of eryptotic RBCs by macrophages [32,112]. There is accumulating evidence generated in animal-based and clinical studies that eryptosis contributes to anemia in liver diseases. In a murine model of bile duct ligation, enhanced RBC turnover is linked to eryptosis [75]. In patients with liver diseases, reduced RBC count correlated with eryptosis and bilirubin levels, suggesting that eryptosis-associated anemia was linked to the elevated bilirubin plasma concentrations [75]. Furthermore, in patients with HB-ACLF, eryptosis parameters including Ca^2+^ and ROS levels negatively correlated with the hemoglobin content in RBCs, erythrocyte count, and hematocrit [98]. Thus, anemia in HB-ACLF is directly linked to eryptosis. Zheng et al. demonstrated that anemia in ALD was associated with excessive hemolysis triggered by ethanol directly. In addition, ethanol triggered eryptosis in vitro, which allowed the authors to link ALD-associated anemia with eryptosis [100]. However, there was no direct confirmation of eryptosis activation in heavy drinkers or patients with ALD. Likewise, anemia in high-fat diets, a risk factor for NAFLD, was linked to accelerated eryptosis [102]. However, more clinical studies are necessary to uncover the interplay between anemia and eryptosis in ALD, NAFLD, and other liver diseases.

Our analysis raises awareness of the interplay between anemia and eryptosis in liver diseases and supports studies that might further unravel the role of eryptosis in anemia in liver diseases.

### 5.5. Eryptosis in Liver Diseases Promotes Blood Coagulation

Accumulating evidence suggests that besides anemia, eryptosis in liver diseases can promote coagulation. For instance, erythrocytes derived from cirrhotic patients promoted fibrin formation and decreased the coagulation time, clearly indicating activation of coagulation [99]. Importantly, in obstructive jaundice accompanied by the elevation of unconjugated bilirubin in blood, PS-positive platelets, neutrophils, and circulating microparticles showed procoagulant activity [113]. Taken together, this might indicate that prothrombotic effects of eryptotic cells are associated with PS exposure. Although there is some evidence outlined above that PS-displaying erythrocytes confer hypercoagulation in liver cirrhosis, more high-quality in vivo studies are necessary to strengthen our understanding of the cause-and-effect relationships between eryptosis and thrombosis in liver diseases. In addition, it remains obscure whether eryptosis-mediated hypercoagulation might be promoted by other signaling mechanisms besides PS.

### 5.6. Eryptotic Erythrocytes Trigger Ferroptosis in Hepatic Diseases

A growing body of evidence indicates that phagocytosis of erythrocytes by macrophages promotes ferroptosis, a lipid peroxidation-dependent and Fe^2+^-mediated type of RCD [114,115,116]. It is important to note that PS-dependent erythrophagocytosis triggers ferroptosis of endothelial cells, promoting endothelial dysfunction and blood coagulation [117]. Contribution of erythrophagocytosis-induced ferroptosis to the pathogenesis of atherosclerosis and renal diseases has been already demonstrated [115,118,119]. Since PS-mediated mechanisms are critical for erythrophagocytosis, it can be suggested that ferroptosis induction can be another mechanism responsible for eryptosis-related detrimental effects. It has been hypothesized that eryptotic erythrocytes might trigger ferroptosis in liver diseases. Indeed, using an animal model of NASH, Park et al. showed that phagocytosis of PS-exposing RBCs triggered ferroptosis of immune cells and hepatocytes. Notably, this process could be mitigated by cilostazol, a phosphodiesterase 3A and phosphodiesterase 3B inhibitor [97]. At the same time, in MASLD, erythrophagocytosis-associated ferroptosis was not mediated by PS exposure [120], suggesting that eryptosis is just one of the several mechanisms that can promote ferroptosis through phagocytosis of RBCs.

Thus, ferroptosis induced by efferocytosis of eryptotic cells might contribute to vascular thrombosis, microvascular remodeling, and inflammation enhancement in liver diseases. Further investigation of this topic might unlock novel therapeutic opportunities for therapy of liver diseases.

### 5.7. Erythrocytes Are Also Cleared by PS-Independent Pathways in Liver Diseases

PS exposure is a potent incentive for erythrocytes to be engulfed by phagocytes [34]. However, erythrocytes might be cleared from the blood through alternative pathways, e.g., senescent erythrocytes accumulate the band 3 protein-derived senescent erythrocyte-specific antigen (SESA), which binds to autologous antibodies promoting erythrophagocytosis [121]. Furthermore, erythrocytes lose anti-phagocytic CD47 [122] and/or sialic acids [123] and accumulate sphingolipids [124] with aging, which results in accelerated phagocytosis of senescent erythrocytes. Our analysis has revealed that PS-independent clearance of erythrocytes is observed in liver diseases. For instance, in patients with NAFLD, erythrocytes experience CD47 depletion and sphingosine overload, simultaneously releasing monocyte chemoattractant protein-1 (MCP1). This ensures erythrophagocytosis [125]. Indeed, liver infiltration with monocytes in NAFLD has been attributed to the release of chemokines by erythrocytes [126]. Notably, hepatic macrophages are influenced by phagocytosis of hemolytic erythrocytes, which switches their phenotype to the anti-inflammatory one [127]. Spur-cell anemia is frequently observed in patients with ALD and is associated with excessive hemolysis [128]. In spur-cell anemia, membranes of erythrocytes have higher levels of cholesterol [129], and excessive cholesterol in cell membranes of RBCs is known to prevent PS translocation to the outer leaflet [130]. The role of eryptosis in spur-cell anemia in ALD is poorly understood, but given that Ca^2+^ chelating agents are effective in this case, Ca^2+^-dependent eryptosis may be involved [131]. Furthermore, eryptosis is known to be associated with changes in lipid membranes, including modifications of the fluidity and lipid order [35,132]. In particular, lipid order in phospholipid bilayers of cell membranes in eryptotic erythrocytes can be either increased or decreased [132,133]. At the same time, there is compelling evidence that erythrocyte membrane fluidity is altered in liver diseases. Owen et al. reported that erythrocyte membrane fluidity was decreased in liver diseases [134]. Likewise, it was found to be reduced in ALD [135]. Conversely, in alcohol-induced liver cirrhosis, erythrocyte membrane fluidity was reported to be increased [136]. There is some evidence that changes in erythrocyte membrane fluidity might affect the ability of phagocytes to engulf erythrocytes [137], indicating that membrane fluidity might be implicated in the clearance of erythrocytes in liver diseases.

To sum up, besides efferocytosis of eryptotic PS-displaying erythrocytes, there are alternative erythrocyte removal pathways that might significantly affect the immune response in liver diseases.

### 5.8. Eryptosis Is an Important Factor in Hepatic Diseases

Although much progress has been made in our comprehension of the pathological role of eryptosis, especially in renal diseases [138], its significance in liver diseases remains poorly investigated. There is growing consensus that accelerated eryptosis might cause anemia and thrombosis in liver diseases, and PS has long been recognized as a key factor in this case. However, the specific details of this process should be clarified in further studies. Moreover, eryptosis indices might show promise as diagnostic and prognostic markers in liver diseases. To the best of our knowledge, diagnostic and prognostic significance of eryptosis parameters has not been elucidated in liver diseases.

## 6. Eryptosis and Drugs Used for the Treatment of Liver Diseases

Hematological toxicity is a common complication of liver disease medications. In this section, we highlight the ability of drugs used to treat liver diseases to induce eryptosis. This might provide insights into the mechanisms of drug-associated anemia in patients with liver diseases and improve the management of these patients. Unexpectedly, our analysis revealed that the ability of the drugs approved by the U.S. Food and Drug Administration (FDA) to induce eryptosis was poorly studied. This might result in the underestimation of the role of eryptosis as a contributor to anemia development as a side effect liver diseases treatment. For instance, sorafenib, which is a widely used and FDA-approved drug in hepatocellular carcinoma, triggered eryptosis associated with PS exposure, Ca^2+^ overload, and ROS abundance. However, sorafenib-induced eryptosis was ceramide independent and was not linked to ATP depletion [139]. Ca^2+^ elevation triggered PS externalization in erythrocytes in response to ribavirin, a potent FDA-approved antiviral agent administered in HCV infection [140]. Nevertheless, triggers of Ca^2+^ overload in this study remained obscure.

As summarized in Table 3, a variety of liver disease medications are capable of triggering eryptosis [139,140,141,142,143,144,145]. However, apart from sorafenib and ribavirin, the majority of them have not been granted FDA approval. Since NAFLD is associated with fat deposition in hepatocytes, unsaturated fatty acids are considered beneficial in this pathology as a dietary supplement that improves lipid metabolism [146]. Linolenic acid, a polyunsaturated fatty acid (PUFA), triggered PS externalization through oxidative stress and Ca^2+^-dependent mechanisms [145]. Importantly, CK1α activation was required for execution of linolenic acid-induced eryptosis. However, the erythrotoxicity of linolenic acid could be attributable not only to eryptosis. Alharthy et al. demonstrated that linolenic acid-induced RBC death was linked to the activation of MLKL [145], a key enzyme of necroptosis (programmed necrosis of erythrocytes) [147]. Eicosapentaenoic and docosahexaenoic acids are other PUFAs that have hepatoprotective effects in NAFLD [148,149,150,151]. Like linolenic acid, eicosapentaenoic acid promoted eryptosis in a Ca^2+^-dependent manner. At the same time, oxidative stress was not implicated in eicosapentaenoic acid-induced eryptosis [143]. In contrast to eicosapentaenoic acid, docosahexaenoic acid-induced Ca^2+^-dependent eryptosis relied on ROS and Rac1 GTPase signaling [142]. Thus, eryptosis triggered by PUFAs is characterized by diverse signaling pathways involved. However, all of them culminate in Ca^2+^ overload and subsequent PS exposure.

In multiple studies, emodin, a Chinese herb-derived anthraquinone, has been reported to elicit hepatoprotective effects, mitigating hepatic inflammation and injury [152,153]. Emodin triggered eryptosis via promoting Ca^2+^ signaling, oxidative stress, and ceramide formation [144]. Again, our analysis underscores the importance of Ca^2+^ signaling in eryptosis. Notably, other drugs might induce eryptosis in a Ca^2+^-independent manner. In particular, cyclosporine, an immunosuppressive drug used after liver transplantation [154], did not increase the intracellular Ca^2+^ levels. On the contrary, they were unexpectedly decreased. However, cyclosporine promoted accumulation of ceramide and reduced ATP, which resulted in PS exposure [141].

Interestingly, a plant-derived Wnt pathway inhibitor, wogonin, whose hepatoprotective effects are well documented [155,156,157], prevented eryptosis [158], suggesting its potential application in the complex therapy of liver diseases to mitigate the pro-eryptotic effects of other liver disease medications. It seems promising to identify hepatoprotective compounds with anti-eryptotic effects that can be applied in the complex treatment of liver diseases to prevent eryptosis-associated pathological changes.

Although our study bridges the gap between eryptosis and anemia as a complication of the therapy of liver diseases, more studies are required to link eryptosis and drug-induced anemia in patients treated for liver diseases. It is important to note that FDA-approved drugs are primary candidates to be evaluated.

## 7. Concluding Remarks and Challenges

The discovery of RCD pathways such as eryptosis [33,34,55,159] and necroptosis [147,160] in erythrocytes points to a paradigmatic shift in our understanding of fundamental RBC biology. Eryptosis has been brought to the forefront of this research due to the important physiological and pathological roles of RCD. In this review, we have summarized the molecular mechanisms of eryptosis, its role in maintaining a pool of healthy erythrocytes, and possible consequences of accelerated eryptosis. Its emerging role in the pathogenesis of liver diseases has been highlighted to demonstrate that the issue of eryptosis contribution cannot be neglected. Enhanced eryptosis in liver diseases mediates anemia and hypercoagulation, which suggests that eryptosis has a significant impact on the progression of hepatic diseases. Furthermore, this review summarizes that a wide array of endogenous metabolites can act as triggers of eryptosis. Their circulating levels are elevated in liver diseases, exposing erythrocytes to higher concentrations than in normal conditions. In particular, bilirubin and bile acids are currently considered the major contributors to accelerated eryptosis in liver diseases. Likewise, a variety of drugs administered to patients with liver diseases are described as eryptosis inducers, which enhances awareness of eryptosis as a side effect of the therapy. Recent advances in the field provide an optimistic view of the perspectives to uncover the role of eryptosis in liver diseases, but more studies are required to obtain the full picture. A wide range of answers to unresolved questions should be unveiled to completely decipher the intricate role of eryptosis in liver diseases. In particular, to the best of our knowledge, there is an extremely limited set of liver diseases in which eryptosis has been investigated, and the data are often conflicting. As clearly outlined in this review, pathological processes triggered by eryptosis like anemia, thrombosis, and endothelial damage are PS dependent. At the same time, PS-independent mechanisms mediating eryptosis-associated alterations are not described. Furthermore, there is some evidence that efferocytosis of eryptotic erythrocytes by Kupffer cells might contribute to fibrosis development in NASH [95]. However, more studies are needed to refine this concept. The diagnostic and prognostic potential of eryptosis indices in individual liver diseases is poorly studied.

Although there are some advances in elucidating the role of eryptosis in liver diseases, translation of the findings into clinical practice seems challenging. Given the strong role of PS externalization in mediating erythrophagocytosis and procoagulant effects and a wide array of upstream signaling pathways resulting in PS exposure (Ca^2+^ signaling, caspase-3, ROS, or ceramide), including in liver diseases, targeting PS exposed on the cell surface rather than the key signaling components of eryptosis seems to be a more effective pharmacological strategy.

## Figures and Tables

**Figure 1 medsci-13-00125-f001:**
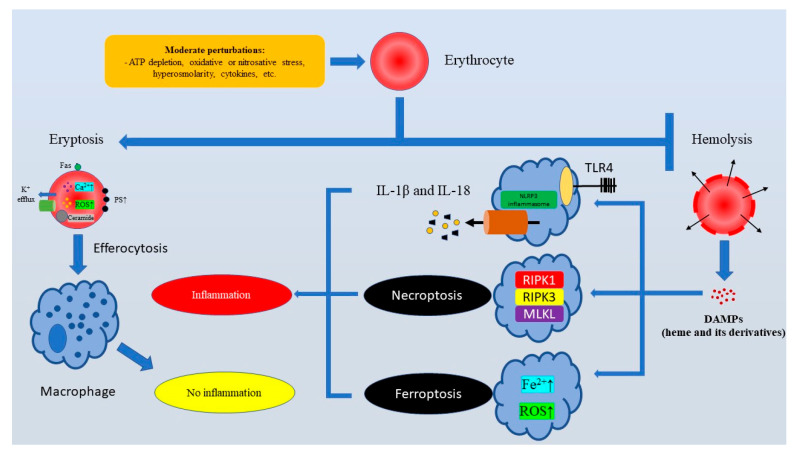
The major physiological function of eryptosis is to ensure the rapid clearance of injured erythrocytes to prevent pro-inflammatory hemolysis. Subhemolytic stress stimuli cause Ca^2+^ overload in erythrocytes and PS externalization. PS exposure is crucial for efferocytosis (PS-mediated erythrophagocytosis). Efferocytosis of eryptotic erythrocytes is “immunologically silent”. At the same time, hemolysis triggers the TLR4/NLRP3-dependent release of pro-inflammatory IL-1β and IL-18, as well as immunogenic cell deaths like ferroptosis and necroptosis. Abbreviations: ATP, Adenosine triphosphate; DAMPs, Damage-associated molecular patterns; IL-18, Interleukin 18; IL-1β, Interleukin 1 beta; MLKL, Mixed lineage kinase domain like pseudokinase; NLRP3, NLR Family Pyrin Domain-Containing 3; PS, Phosphatidylserine; RIPK1, Receptor-interacting serine/threonine-protein kinase 1; RIPK3, Receptor-interacting serine/threonine-protein kinase 3; ROS, Reactive oxygen species; TLR4, Toll-like receptor 4.

**Figure 2 medsci-13-00125-f002:**
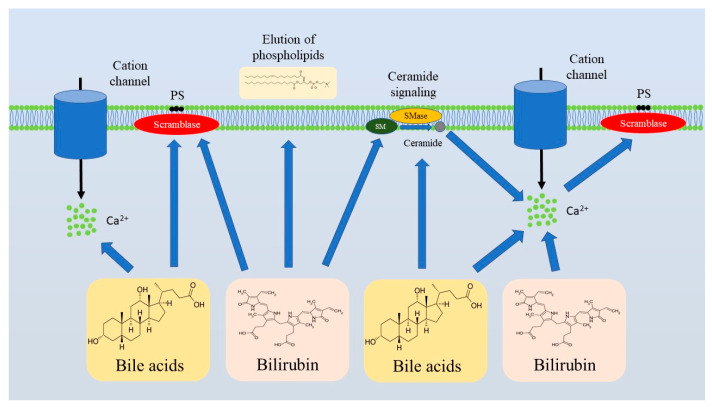
Bilirubin and bile acids are the major triggers of eryptosis in liver diseases. Bilirubin and bile acids induce eryptosis by promoting ceramide formation and Ca^2+^ influx that culminate in PS externalization. Abbreviations: PS, Phosphatidylserine; SM, Sphingomyelin; SMase, Sphingomyelinase.

**Table 1 medsci-13-00125-t001:** Liver-associated metabolites or proteins as inducers of erythrocyte cell death.

Metabolite	Description	Dose	Signaling and Cell Death Features	Reference
Chenodeoxycholic acid (unconjugated, taurine and glycine conjugates)	A bile acid	0.3 mM	Ca^2+^ overload, hemolysis	[74]
Cholic acid (unconjugated, taurine and glycine conjugates)	A bile acid	0.3 mM	Ca^2+^ overload, hemolysis	[74]
Conjugated bilirubin	A heme breakdown product conjugated with glucuronic acid	Above 3 mg/dL	Ca^2+^- and ceramide-dependent eryptosis	[75]
CRP	An acute phase protein	5 µg/mL and above	Ca^2+^-, caspase-3-, and ceramide-dependent eryptosis	[76]
Deoxycholic acid (unconjugated, taurine and glycine conjugates)	A bile acid	0.3 mM	Ca^2+^ overload, hemolysis	[74]
Glycochenodesoxycholic acid	A bile acid	500 µM and above	Ca^2+^- and ceramide-dependent eryptosis	[77]
Lauric acid	A saturated fatty acid	100 µM and above	Ca^2+^-, CK1α-, and oxidative stress-dependent eryptosis	[78]
Lithocholic acid (unconjugated, taurine and glycine conjugates)	A bile acid	0.3 mM	Ca^2+^ overload, hemolysis	[74]
Taurochenodesoxycholic acid	A bile acid	125 µM and above	Ca^2+^- and ceramide-dependent eryptosis	[77]
Unconjugated bilirubin	A heme breakdown product non-conjugated with glucuronic acid	Bilirubin/albumin ratio of 1 and above	PS externalization, elution of phospholipid from cell membranes, formation of echinocytes, hemolysis	[79,80]

Abbreviations: CK1α, Casein kinase 1α; CRP, C-reactive protein; PS, Phosphatidylserine.

**Table 2 medsci-13-00125-t002:** Eryptosis in liver diseases.

Disease	Signaling	Reference	Notes
NASH	N/A	[95]	A rabbit model: Eryptotic erythrocytes are phagocyted by Kupffer cells
NASH	Fas-dependent eryptosis	[97]	A murine model
Unspecified liver diseases	N/A	[75]	A murine model of bile duct ligation: Eryptosis induction correlates with bilirubin levels
Hepatitis B-related acute-on-chronic liver failure	ROS-dependent eryptosis	[98]	-
Liver cirrhosis	N/A	[99]	Severity-dependent degree of PS exposure

Abbreviations: N/A, Not available; NASH, Non-alcoholic steatohepatitis; PS, Phosphatidylserine; ROS, Reactive oxygen species.

**Table 3 medsci-13-00125-t003:** Eryptosis-inducing drugs used to treat liver diseases.

Drug	Description	Dose	Signaling	Reference
Ribavirin	A guanosine analogue approved by the FDA for the treatment of HCV infection	8 µg/mL and above	Ca^2+^-dependent eryptosis	[140]
Sorafenib	A multikinase inhibitor approved by the FDA for the treatment of hepatocellular carcinoma	0.5 µM and above	Ca^2+^- and ROS-dependent eryptosis	[139]
Cyclosporine	An immunosuppressive drug administered after liver transplantation	5 µM and above	Ceramide-dependent eryptosis	[141]
Docosahexaenoic acid	A polyunsaturated fatty acid used for the treatment of NAFLD	80 µM and above	Ca^2+^-, ROS-, and Rac1 GTPase-dependent eryptosis	[142]
Eicosapentaenoic acid	A polyunsaturated fatty acid used for the treatment of NAFLD	20 µM and above	Ca^2+^-dependent eryptosis	[143]
Emodin	A component of several Chinese herbs with hepatoprotective effects	10 µM and above	Ca^2+^-, ceramide-, and ROS-dependent eryptosis	[144]
Linolenic acid	A polyunsaturated fatty acid used for the treatment of NAFLD	80 µM and above	Ca^2+^-, ROS-, CK1α-, and MLKL-dependent cell death	[145]

Abbreviations: CK1α, Casein kinase 1α; FDA, U.S. Food and Drug Administration; HCV, Hepatitis C virus; MLKL, Mixed lineage kinase domain-like pseudokinase; NAFLD, Non-alcoholic fatty liver disease; ROS, Reactive oxygen species.

## Data Availability

The data that support this study are available from the corresponding authors AT and KB upon reasonable request.

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
