# Peer review of "Eryptosis in Liver Diseases: Contribution to Anemia and Hypercoagulation"

_medsci, 2025, doi:10.3390/medsci13030125_

Round 1

Reviewer 1 Report

Comments and Suggestions for Authors

There are some references that i feel the authors should have cited. 

Park JB, Ko K, Baek YH, Kwon WY, Suh S, Han SH, Kim YH, Kim HY, Yoo YH. Pharmacological Prevention of Ectopic Erythrophagocytosis by Cilostazol Mitigates Ferroptosis in NASH. Int J Mol Sci. 2023 Aug 16;24(16):12862. doi: 10.3390/ijms241612862. PMID: 37629045; PMCID: PMC10454295.

Kyriakou Z, Mimidis K, Politis N, Veniamis P, Vlachos D, Anagnostopoulos K, Papadopoulos C. Reduced Erythrocyte Opsonization by Calreticulin, Lactadherin, Mannose-binding Lectin, and Thrombospondin-1 in MAFLD Patients. Cardiovasc Hematol Disord Drug Targets. 2025 Jun 24. doi: 10.2174/011871529X381576250613041457. Epub ahead of print. PMID: 40600531.

Charalampos P. The Molecular Determinants of Erythrocyte Removal Impact the Development of Metabolic Dysfunction-Associated Steatohepatitis. Endocr Metab Immune Disord Drug Targets. 2024 Dec 9. doi: 10.2174/0118715303362972241121062515. Epub ahead of print. PMID: 39660491.

Pfefferlé M, Ingoglia G, Schaer CA, Yalamanoglu A, Buzzi R, Dubach IL, Tan G, López-Cano EY, Schulthess N, Hansen K, Humar R, Schaer DJ, Vallelian F. Hemolysis transforms liver macrophages into antiinflammatory erythrophagocytes. J Clin Invest. 2020 Oct 1;130(10):5576-5590. doi: 10.1172/JCI137282. PMID: 32663195; PMCID: PMC7524492.

I would suggest also that the authors need to add a small sections of alternative erythrocyte removal pathways in liver diseases, such as CD47 reduction, sphingosine accumulation and chemokine release.

Dupuis L, Chauvet M, Bourdelier E, Dussiot M, Belmatoug N, Le Van Kim C, Chêne A, Franco M. Phagocytosis of Erythrocytes from Gaucher Patients Induces Phenotypic Modifications in Macrophages, Driving Them toward Gaucher Cells. Int J Mol Sci. 2022 Jul 11;23(14):7640. doi: 10.3390/ijms23147640. PMID: 35886988; PMCID: PMC9319206.

Papadopoulos C, Spourita E, Mimidis K, Kolios G, Tentes L, Anagnostopoulos K. Nonalcoholic Fatty Liver Disease Patients Exhibit Reduced CD47 and Increased Sphingosine, Cholesterol, and Monocyte Chemoattractant Protein-1 Levels in the Erythrocyte Membranes. Metab Syndr Relat Disord. 2022 Sep;20(7):377-383. doi: 10.1089/met.2022.0006. Epub 2022 May 9. PMID: 35532955.

Papadopoulos C, Mimidis K, Papazoglou D, Kolios G, Tentes I, Anagnostopoulos K. Red Blood Cell-Conditioned Media from Non-Alcoholic Fatty Liver Disease Patients Contain Increased MCP1 and Induce TNF-α Release. Rep Biochem Mol Biol. 2022 Apr;11(1):54-62. doi: 10.52547/rbmb.11.1.54. PMID: 35765536; PMCID: PMC9208556.

In general, this is a very well-written, comprehensive and significant review article, where the authors summarise the mechanisms and implications of eryptosis in liver diseases and their treatments. As i said, i feel however that the authors should comment on the role of alternative erythrocyte removal pathways. Finally, i would suggest that the authors extend the description of the role of erythrocyte removal in ferroptosis. For more details see the review cited below, and references therein.

Papadopoulos C. Molecular and Immunometabolic Landscape of Erythrophagocytosis-induced Ferroptosis. Cardiovasc Hematol Disord Drug Targets. 2025 Apr 14. doi: 10.2174/011871529X370553250322095430. Epub ahead of print. PMID: 40231500.

Author Response

There are some references that i feel the authors should have cited. 

Park JB, Ko K, Baek YH, Kwon WY, Suh S, Han SH, Kim YH, Kim HY, Yoo YH. Pharmacological Prevention of Ectopic Erythrophagocytosis by Cilostazol Mitigates Ferroptosis in NASH. Int J Mol Sci. 2023 Aug 16;24(16):12862. doi: 10.3390/ijms241612862. PMID: 37629045; PMCID: PMC10454295.

Thank you. The reference was cited and the findings of this study were discussed in Subsection 5.6. Eryptotic erythrocytes trigger ferroptosis in NASH.

Kyriakou Z, Mimidis K, Politis N, Veniamis P, Vlachos D, Anagnostopoulos K, Papadopoulos C. Reduced Erythrocyte Opsonization by Calreticulin, Lactadherin, Mannose-binding Lectin, and Thrombospondin-1 in MAFLD Patients. Cardiovasc Hematol Disord Drug Targets. 2025 Jun 24. doi: 10.2174/011871529X381576250613041457. Epub ahead of print. PMID: 40600531.

Thank you. The reference was cited and the findings of this study were discussed in Subsection 5.6. Eryptotic erythrocytes trigger ferroptosis in NASH.

Charalampos P. The Molecular Determinants of Erythrocyte Removal Impact the Development of Metabolic Dysfunction-Associated Steatohepatitis. Endocr Metab Immune Disord Drug Targets. 2024 Dec 9. doi: 10.2174/0118715303362972241121062515. Epub ahead of print. PMID: 39660491.

Thank you. The reference was cited and the findings of this study were discussed in Subsection 5.6. Eryptotic erythrocytes trigger ferroptosis in NASH.

Pfefferlé M, Ingoglia G, Schaer CA, Yalamanoglu A, Buzzi R, Dubach IL, Tan G, López-Cano EY, Schulthess N, Hansen K, Humar R, Schaer DJ, Vallelian F. Hemolysis transforms liver macrophages into antiinflammatory erythrophagocytes. J Clin Invest. 2020 Oct 1;130(10):5576-5590. doi: 10.1172/JCI137282. PMID: 32663195; PMCID: PMC7524492.

Thank you. The reference was cited and the findings of this study were discussed in Subsection 5.7. Erythrocytes are also cleared by PS-independent pathways in liver diseases.

I would suggest also that the authors need to add a small sections of alternative erythrocyte removal pathways in liver diseases, such as CD47 reduction, sphingosine accumulation and chemokine release.

Thank you for pointing this out. Subsection 5.7. Erythrocytes are also cleared by PS-independent pathways in liver diseases was added (Lines 472-505).

PS exposure is a potent incentive for erythrocytes to be engulfed by phagocytes [34]. However, erythrocytes might be cleared from the blood through alternative pathways, e.g., senescent erythrocytes accumulate the band 3 protein-derived senescent erythrocyte-specific antigen (SESA), which binds to autologous antibodies promoting erythrophagocytosis [121]. Furthermore, erythrocytes lose anti-phagocytic CD47 [122] and/or sialic acids [123] and accumulate sphingolipids [124] with aging, which results in accelerated phagocytosis of senescent erythrocytes. Our analysis has revealed that PS-independent clearance of erythrocytes is observed in liver diseases. For instance, in patients with NAFLD, erythrocytes experience CD47 depletion and sphingosine overload simultaneously releasing monocyte chemoattractant protein-1 (MCP1). This ensures erythrophagocytosis [125]. Indeed, liver infiltration with monocytes in NAFLD has been attributed to the release of chemokines by erythrocytes [126]. Notably, hepatic macrophages are influenced by phagocytosis of hemolytic erythrocytes, which switches their phenotype to the anti-inflammatory one [127]. Spur-cell anemia is frequently observed in patients with ALD and is associated with excessive hemolysis [128]. In spur-cell anemia, membranes of erythrocytes have higher levels of cholesterol [129] and excessive cholesterol in cell membranes of RBCs is known to prevent PS translocation to the outer leaflet [130]. The role of eryptosis in spur-cell anemia in ALD is poorly understood, but given that Ca2+ chelating agents are effective in this case, Ca2+-dependent eryptosis may be involved [131]. Furthermore, eryptosis is known to be associated with changes in lipid membranes, including modifications of the fluidity and lipid order [35,132]. In particular, lipid order in phospholipid bilayers of cell membranes in eryptotic erythrocytes can be either increased or decreased [132,133]. At the same time, there is compelling evidence that erythrocyte membrane fluidity is altered in liver diseases. Owen et al. reported that erythrocyte membrane fluidity was decreased in liver diseases [134]. Likewise, it was found to be reduced in ALD [135]. Conversely, in alcohol-induced liver cirrhosis, erythrocyte membrane fluidity was reported to be increased [136]. There is some evidence that changes in erythrocyte membrane fluidity might affect the ability of phagocytes to engulf erythrocytes [137], indicating that the membrane fluidity might be implicated in the clearance of erythrocytes in liver diseases.

To sum up, besides efferocytosis of eryptotic PS-displaying erythrocytes, there are alternative erythrocyte removal pathways, which might significantly affect the immune response in liver diseases.

Dupuis L, Chauvet M, Bourdelier E, Dussiot M, Belmatoug N, Le Van Kim C, Chêne A, Franco M. Phagocytosis of Erythrocytes from Gaucher Patients Induces Phenotypic Modifications in Macrophages, Driving Them toward Gaucher Cells. Int J Mol Sci. 2022 Jul 11;23(14):7640. doi: 10.3390/ijms23147640. PMID: 35886988; PMCID: PMC9319206.

Thank you. The reference was cited and the findings of this study were discussed in Subsection 5.7. Erythrocytes are also cleared by PS-independent pathways in liver diseases.

Papadopoulos C, Spourita E, Mimidis K, Kolios G, Tentes L, Anagnostopoulos K. Nonalcoholic Fatty Liver Disease Patients Exhibit Reduced CD47 and Increased Sphingosine, Cholesterol, and Monocyte Chemoattractant Protein-1 Levels in the Erythrocyte Membranes. Metab Syndr Relat Disord. 2022 Sep;20(7):377-383. doi: 10.1089/met.2022.0006. Epub 2022 May 9. PMID: 35532955.

Thank you. The reference was cited and the findings of this study were discussed in Subsection 5.7. Erythrocytes are also cleared by PS-independent pathways in liver diseases.

Papadopoulos C, Mimidis K, Papazoglou D, Kolios G, Tentes I, Anagnostopoulos K. Red Blood Cell-Conditioned Media from Non-Alcoholic Fatty Liver Disease Patients Contain Increased MCP1 and Induce TNF-α Release. Rep Biochem Mol Biol. 2022 Apr;11(1):54-62. doi: 10.52547/rbmb.11.1.54. PMID: 35765536; PMCID: PMC9208556.

Thank you. The reference was cited and the findings of this study were discussed in Subsection 5.7. Erythrocytes are also cleared by PS-independent pathways in liver diseases.

In general, this is a very well-written, comprehensive and significant review article, where the authors summarise the mechanisms and implications of eryptosis in liver diseases and their treatments. As i said, i feel however that the authors should comment on the role of alternative erythrocyte removal pathways. Finally, i would suggest that the authors extend the description of the role of erythrocyte removal in ferroptosis. For more details see the review cited below, and references therein.

Thank you for the positive assessment of our manuscript. We really appreciate your suggestion to highlight the role of eryptosis in ferroptosis induction. The corresponding subsection was added (Lines 451-471).

5.6. Eryptotic erythrocytes trigger ferroptosis in hepatic diseases

A growing body of evidence indicates that phagocytosis of erythrocytes by macrophages promotes ferroptosis, a lipid peroxidation-dependent and Fe2+-mediated type of RCD [114-116]. It is important to note that PS-dependent erythrophagocytosis triggers ferroptosis of endothelial cells, promoting endothelial dysfunction and blood coagulation [117]. Contribution of erythrophagocytosis-induced ferroptosis to the pathogenesis of atherosclerosis and renal diseases has been already demonstrated [115,118,119]. Since PS-mediated mechanisms are critical for erythrophagocytosis, it can be suggested that ferroptosis induction can be another mechanism responsible for eryptosis-related detrimental effects. It has been hypothesized that eryptotic erythrocytes might trigger ferroptosis in liver diseases. Indeed, using an animal model of NASH, Park et al. showed that phagocytosis of PS-exposing RBCs triggered ferroptosis of immune cells and hepatocytes. Notably, this process could be mitigated by cilostazol, a phosphodiesterase 3A and phosphodiesterase 3B inhibitor [97]. At the same time, in MASLD, erythrophagocytosis-associated ferroptosis was not mediated by PS exposure [120], suggesting that eryptosis is just one of the several mechanisms that can promote ferroptosis through phagocytosis of RBCs.

Thus, ferroptosis induced by efferocytosis of eryptotic cells might contribute to vascular thrombosis, microvascular remodeling, and inflammation enhancement in liver diseases. Further investigation of this topic might unlock novel therapeutic opportunities for therapy of liver diseases.

Papadopoulos C. Molecular and Immunometabolic Landscape of Erythrophagocytosis-induced Ferroptosis. Cardiovasc Hematol Disord Drug Targets. 2025 Apr 14. doi: 10.2174/011871529X370553250322095430. Epub ahead of print. PMID: 40231500.

Thank you. The reference was cited and the findings of this study were discussed in Subsection 5.6. Eryptotic erythrocytes trigger ferroptosis in NASH.

We want to express our sincere gratitude to the Reviewer for the improvements of our manuscript. Our modifications were performed in the Track Changes mode.

Reviewer 2 Report

Comments and Suggestions for Authors

Authors reviewed about eryptosis and its association with liver diseases. Little has been summarized in such topic, This review was interesting. Several issues remained to be addressed.

  1. In the introduction section, authors described some liver diseases. MASLD and NAFLD were almost same disease. They should not be written side by side.
  2. In chronic liver diseases, especially in cirrhosis, macrocytic anemia is usually found. Some disorder in erythrocyte membrane is found. Furthermore, spur-cell anemia is known in alcoholic liver failure. Authors should describe the association between eryptosis and such erythrocyte membrane disorders.
  3. In Table 3, authors showed some drugs to be used to treat liver disease. But part of them (below half) is not commonly used or is not approved for the treatment of liver diseases. It should be clarified or discussed in the manuscript.

Author Response

Authors reviewed about eryptosis and its association with liver diseases. Little has been summarized in such topic, This review was interesting. Several issues remained to be addressed.

Thank you for your positive feedback and evaluation of our manuscript. We really appreciate your comments.

  1. In the introduction section, authors described some liver diseases. MASLD and NAFLD were almost same disease. They should not be written side by side.

Thank you for pointing this out. Modified accordingly (Lines 70-77):

In particular, chronic liver disease (CLD), which encompasses a long-lasting hepatic inflammation (over 6 months) resulting in damage and altered regeneration of the parenchyma, eventually culminating in fibrosis and cirrhosis [13], comprises NAFLD currently known as metabolic dysfunction-associated steatotic liver disease (MASLD) [14], alcohol-associated liver disease (ALD), chronic viral hepatitis, autoimmune hepatitis, genetic disorders (alpha-1 antitrypsin deficiency, hereditary hemochromatosis, or Wilson disease), or drug-induced liver injury [15,16].

  1. In chronic liver diseases, especially in cirrhosis, macrocytic anemia is usually found. Some disorder in erythrocyte membrane is found. Furthermore, spur-cell anemia is known in alcoholic liver failure. Authors should describe the association between eryptosis and such erythrocyte membrane disorders.

Thank you. We have added this discussion to the following subsection (Lines 472-505):

PS exposure is a potent incentive for erythrocytes to be engulfed by phagocytes [34]. However, erythrocytes might be cleared from the blood through alternative pathways, e.g., senescent erythrocytes accumulate the band 3 protein-derived senescent erythrocyte-specific antigen (SESA), which binds to autologous antibodies promoting erythrophagocytosis [121]. Furthermore, erythrocytes lose anti-phagocytic CD47 [122] and/or sialic acids [123] and accumulate sphingolipids [124] with aging, which results in accelerated phagocytosis of senescent erythrocytes. Our analysis has revealed that PS-independent clearance of erythrocytes is observed in liver diseases. For instance, in patients with NAFLD, erythrocytes experience CD47 depletion and sphingosine overload simultaneously releasing monocyte chemoattractant protein-1 (MCP1). This ensures erythrophagocytosis [125]. Indeed, liver infiltration with monocytes in NAFLD has been attributed to the release of chemokines by erythrocytes [126]. Notably, hepatic macrophages are influenced by phagocytosis of hemolytic erythrocytes, which switches their phenotype to the anti-inflammatory one [127]. Spur-cell anemia is frequently observed in patients with ALD and is associated with excessive hemolysis [128]. In spur-cell anemia, membranes of erythrocytes have higher levels of cholesterol [129] and excessive cholesterol in cell membranes of RBCs is known to prevent PS translocation to the outer leaflet [130]. The role of eryptosis in spur-cell anemia in ALD is poorly understood, but given that Ca2+ chelating agents are effective in this case, Ca2+-dependent eryptosis may be involved [131]. Furthermore, eryptosis is known to be associated with changes in lipid membranes, including modifications of the fluidity and lipid order [35,132]. In particular, lipid order in phospholipid bilayers of cell membranes in eryptotic erythrocytes can be either increased or decreased [132,133]. At the same time, there is compelling evidence that erythrocyte membrane fluidity is altered in liver diseases. Owen et al. reported that erythrocyte membrane fluidity was decreased in liver diseases [134]. Likewise, it was found to be reduced in ALD [135]. Conversely, in alcohol-induced liver cirrhosis, erythrocyte membrane fluidity was reported to be increased [136]. There is some evidence that changes in erythrocyte membrane fluidity might affect the ability of phagocytes to engulf erythrocytes [137], indicating that the membrane fluidity might be implicated in the clearance of erythrocytes in liver diseases.

To sum up, besides efferocytosis of eryptotic PS-displaying erythrocytes, there are alternative erythrocyte removal pathways, which might significantly affect the immune response in liver diseases.

  1. In Table 3, authors showed some drugs to be used to treat liver disease. But part of them (below half) is not commonly used or is not approved for the treatment of liver diseases. It should be clarified or discussed in the manuscript.

Thank you for pointing this out. We have mentioned that in the text. However, to emphasize this once again, the following sentence was added (Lines 538-539).

However, apart from sorafenib and ribavirin, the majority of them are not granted with FDA approval.

We want to express our sincere gratitude to the Reviewer for the improvements of our manuscript. Our modifications were performed in the Track Changes mode.

Reviewer 3 Report

Comments and Suggestions for Authors

The paper „Eryptosis in liver diseases: contribution to anemia and hypercoagulation“ gives an expert opinion based on the review of the current literature about the contribution of anemia and hypercoagulation to the exacerbation of liver disease. It is a novel and original approach that discusses an important medical topic due to the clinical experience and scientific outlook of the authors. The paper is worth of considering for publication. Some minor points need to be addressed. Compliments to the authors for the well-organized tables and informative and visually clear figures.

  1. Even though the paper is review and it is state-of-art some methodology should be added as which scientific bases were screened based on which key words. The methodology applied for the review process should be given: the scientific base that was searched, the time interval that was considered, the language (English or/and some other), key words applied for the research etc...
  2. The abstract should be more rewritten in order to be structured pointing out the methodology, the results and the conclusion
  3. At the end of the Introduction a clearly defined aim should be set.
  4. Lines 301-305, the influence of oxidative stress on erypthosis should be addressed more carefully. And vise versa does erypthosis aggravates the consequences of oxidative stress leading to circulus vitousus.
  5. Lines 383-394, are there evidence of the anemia and/or hypercoagulation interference of the progression of ALD and or NAFLD. Please expand this part in order to emphasize the direct influence of eryptosis in liver diseases as required by the Title of the manuscript.
  6. Lines 511-512 in conclusion do not belong to this part as the aim of the study should not be part of the conclusion.

Author Response

The paper „Eryptosis in liver diseases: contribution to anemia and hypercoagulation“ gives an expert opinion based on the review of the current literature about the contribution of anemia and hypercoagulation to the exacerbation of liver disease. It is a novel and original approach that discusses an important medical topic due to the clinical experience and scientific outlook of the authors. The paper is worth of considering for publication. Some minor points need to be addressed. Compliments to the authors for the well-organized tables and informative and visually clear figures.

Thank you for your positive feedback and helpful comments. We have tried to address all the mentioned comments.

  1. Even though the paper is review and it is state-of-art some methodology should be added as which scientific bases were screened based on which key words. The methodology applied for the review process should be given: the scientific base that was searched, the time interval that was considered, the language (English or/and some other), key words applied for the research etc...

Thank you for pointing this out. The following sentence was added (Lines 121-129):

Herein, we systemically analyzed the available PubMed-indexed English-language publications (up to June 2025) using the following combinations of search keywords: “eryptosis”/”erythrocyte apoptosis”/”RBC apoptosis”/“phosphatidylserine exposure”/“phosphatidylserine externalization” AND “hepatic disease”/“liver disease”/”alcohol-associated liver disease OR ALD”/”non-alcoholic fatty liver disease OR NAFLD”/”non-alcoholic steatohepatitis OR NASH”/”chronic liver disease OR CLD”/”metabolic dysfunction-associated steatotic liver disease OR MASLD”/”liver cirrhosis”/”hepatitis”/”liver metabolites”/“liver disease markers”/“liver disease medications”/“liver drugs”.

  1. The abstract should be more rewritten in order to be structured pointing out the methodology, the results and the conclusion

Thank you for pointing this out. The abstract was modified to reflect the review methodology and to emphasize the key findings

Eryptosis is a type of regulated cell death of mature erythrocytes characterized by excessive Ca2+ accumulation followed by phosphatidylserine externalization. Eryptosis facilitates erythrophagocytosis resulting in eradication of damaged erythrocytes, which maintains the population of healthy erythrocytes in blood. Over the recent years, a wide array of diseases has been reported to be linked to accelerated eryptosis, which leads to anemia. A growing number of studies furnish evidence that eryptosis is implicated in the pathogenesis of liver diseases. Herein, we summarize the current knowledge of eryptosis signaling, its physiological role, and the impact of eryptosis to anemia and hypercoagulation. In this article, upon systemically analyzing the PubMed-indexed publications, we also provide a comprehensive overview of the role of eryptosis in the spectrum of hepatic diseases, its contribution to the development of complications in liver pathology, metabolites (bilirubin, bile acids, etc.) that might trigger eryptosis in liver diseases, and eryptosis-inducing liver disease medications. Eryptosis in liver diseases contributes to anemia, hypercoagulation, and endothelial damage (via ferroptosis of endothelial cells). Treatment-associated anemia in liver diseases might be at least partly attributed to drug-induced eryptosis. Ultimately, we analyze the concept of inhibiting eryptosis pharmaceutically to prevent eryptosis-associated anemia and thrombosis in liver diseases.

  1. At the end of the Introduction a clearly defined aim should be set.

Thank you. The following sentence was added (Lines 117-118):

Thus, in this review, we primarily aim to assess the impact of eryptosis in liver diseases, emphasizing its contribution to anemia and hypercoagulation.

  1. Lines 301-305, the influence of oxidative stress on erypthosis should be addressed more carefully. And vise versa does erypthosis aggravates the consequences of oxidative stress leading to circulus vitousus.

Thank you for pointing this out. The following sentence was added (Lines 164-170):

In particular, ROS in erythrocytes are generated as a result of hemoglobin autooxidation, Fe2+-driven Fenton reaction, by NADPH oxidase or xanthine oxidoreductase. Moreover, immune cell-derived exogenous ROS may enter erythrocytes as well. It has been reported that ROS induce eryptosis by promoting Ca2+ influx (cation channel-driven eryptosis), in a caspase-3-dependent way, and ceramide production [51]. It can be assumed that ROS-mediated eryptosis might prevent further generation of ROS by eliminating ROS-producing RBCs.

  1. Lines 383-394, are there evidence of the anemia and/or hypercoagulation interference of the progression of ALD and or NAFLD. Please expand this part in order to emphasize the direct influence of eryptosis in liver diseases as required by the Title of the manuscript.

Thank you for pointing this out. The subsection was expanded (Lines 428-434):

Zheng et al. demonstrated that anemia in ALD was associated with excessive hemolysis triggered by ethanol directly. In addition, ethanol triggered eryptosis in vitro, which allowed the authors to link ALD-associated anemia with eryptosis [100]. However, there was no direct confirmation of eryptosis activation in heavy drinkers or patients with ALD. Likewise, anemia in high-fat diet, a risk factor for NAFLD, was linked to accelerated eryptosis [102]. However, more clinical studies are necessary to uncover the interplay between anemia and eryptosis in ALD, NAFLD, and other liver diseases.

  1. Lines 511-512 in conclusion do not belong to this part as the aim of the study should not be part of the conclusion.

Thank you for pointing this out. This sentence was transferred to the end of the Introduction section (Lines 119-120).

We want to express our sincere gratitude to the Reviewer for the improvements of our manuscript. Our modifications were performed in the Track Changes mode.

Round 2

Reviewer 2 Report

Comments and Suggestions for Authors

Revised manuscript was well-addressed to the reviewer's comments and well-written. This review is informative and interesting.